# Allicin as a Volatile or Nebulisable Antimycotic for the Treatment of Pulmonary Mycoses: In Vitro Studies Using a Lung Flow Test Rig

**DOI:** 10.3390/ijms23126607

**Published:** 2022-06-14

**Authors:** Christina Schier, Jana Foerster (née Reiter), Monika Heupel, Philipp Dörner, Michael Klaas, Wolfgang Schröder, Lothar Rink, Alan J. Slusarenko, Martin C. H. Gruhlke

**Affiliations:** 1Department of Plant Physiology, RWTH Aachen University, 52074 Aachen, Germany; alan.slusarenko@bio3.rwth-aachen.de (A.J.S.); martin.gruhlke@rwth-aachen.de (M.C.H.G.); 2GENAWIF e.V.—Society for Natural Compound and Active Ingredient Research, 52070 Aachen, Germany; jana.foerster@genawif.com; 3Landwirtschaftskammer Rheinland, 50765 Köln-Auweiler, Germany; monika.heupel@lwk.nrw.de; 4Chair of Fluid Mechanics and Institute of Aerodynamics, RWTH Aachen University, 52062 Aachen, Germany; p.doerner@aia.rwth-aachen.de (P.D.); m.klaas@aia.rwth-aachen.de (M.K.); office@aia.rwth-aachen.de (W.S.); 5Institute of Immunology, RWTH Aachen University Hospital, 52074 Aachen, Germany; lrink@ukaachen.de; 6Institute of Applied Microbiology—iAMB, Aachener Biology and Biotechnology—ABBt, RWTH Aachen University, 52074 Aachen, Germany

**Keywords:** allicin, mycosis, lung infection, *Paecilomyces*, antimycotic

## Abstract

Fungal infections of the lung are an increasing problem worldwide and the search for novel therapeutic agents is a current challenge due to emerging resistance to current antimycotics. The volatile defence substance allicin is formed naturally by freshly injured garlic plants and exhibits broad antimicrobial potency. Chemically synthesised allicin was active against selected fungi upon direct contact and via the gas phase at comparable concentrations to the pharmaceutically used antimycotic amphotericin B. We investigated the suppression of fungal growth by allicin vapour and aerosols in vitro in a test rig at air flow conditions mimicking the human lung. The effect of allicin via the gas phase was enhanced by ethanol. Our results suggest that allicin is a potential candidate for development for use in antifungal therapy for lung and upper respiratory tract infections.

## 1. Introduction

Apart from bacteria and viruses, people can also become infected by fungi and the resulting diseases are known as mycoses [1]. For example, the thrush fungus *Candida albicans* can infect the mouth/throat area [2] but can also cause a systemic mycosis spreading throughout the entire body [3]. Mycoses of the skin, hair, fingernails and toenails or mucous membranes are also common [4]. A typical example here is athlete’s foot caused by *Trichophyton rubrum* [*ibid*]. Fungal infections of the lung and upper respiratory tract are frequent and unfortunately often fatal in immunosuppressed or multimorbid patients [5]. By far the best-known lung pathogenic fungus is *Aspergillus fumigatus*, a thermotolerant ascomycete which infects the lung and can form so-called aspergilloma, which are ball-like structures visible in X-rays [6,7].

Furthermore, various other fungal species that can grow at a temperature of 37 °C are also potential lung pathogens particularly for people with reduced immune capacity [8]. The great importance of pulmonary mycoses is also reflected in the current COVID-19 pandemic. In addition to bacterial infections following SARS-CoV-2 virus infection, and leading to pneumonia with an often dramatic course, pathological examination showed an astonishing number of pulmonary mycoses, especially in intubated patients [9,10]. Since the beginning of 2021, reports of COVID-19 associated mucormycosis, an infection caused by fungi of the order Mucorales, have been increasing [11,12,13]. The disease is characterised by an extensive clinical presentation, with the most common forms being rhino-orbital, cerebral and pulmonary mucormycoses. Most cases have been reported in India, where the estimated prevalence is about 70 times higher than the estimated global median prevalence of 0.2 cases per 100,000 persons [14,15]. The infection is extremely aggressive and, with a mortality rate of 49%, highly life-threatening, which is why rapid diagnosis and early treatment are essential [16].

There are different classes of substances that can be used to treat mycoses. These include, for example, the azoles which inhibit the synthesis of the sterol ergosterol, which is essential for fungi [17]. A well-known example of an azole fungicide is fluconazole [18]. However, since azoles are also used extensively in agriculture to combat fungal diseases in plants, and the resistance mechanism is primarily non-specific through the increased expression of exporters (such as ATP-binding cassette-containing [ABC] transporters) [19,20,21], there is a fairly high probability of resistance developing, especially since many of the opportunistic lung pathogenic fungi usually live saprophytically in the soil.

Another common class of antimycotics used in human and veterinary medicine are the so-called polyene antimycotics, with amphotericin B being the most commonly used antimycotic of this substance class. Polyenes bind to ergosterol [22] and thus lead to the formation of pores in the fungal plasma membrane, resulting in a depolarisation of the membrane and leakage of ions [23]. However, amphotericin B has very low water solubility, which makes application by oral or intravenous routes extremely complex. Attempts are being made to counteract this by administering the amphotericin in liposomes or with special, novel derivatives [24,25]. Additionally, amphotericin B has a rather high nephrotoxicity, which makes its application even more difficult and to a certain extent risky [26].

Thus, there is a need for novel antimycotics, and one potential candidate is allicin (diallyl thiosulfinate), a sulfur-containing defence substance from the garlic plant (*Allium sativum*) [27,28]. Allicin is formed from the amino acid alliin (*S*-allyl-L-cysteine sulfoxide) by action of the C-S lyase alliinase after injury to the plant tissue. Enzymatic cleavage of alliin releases allyl sulfenic acid, and subsequent spontaneous condensation of two allyl sulfenic acid molecules builds allicin [*ibid*]. Allicin was first described and investigated in terms of its biological functions in 1944 [29], although the antimicrobial properties of the garlic plant were known and described considerably earlier, for example by Louis Pasteur [30].

Allicin is a potent thiol trapping reagent and because of this it has an amazing spectrum of antimicrobial and biocidal functions. In addition to its effect against phyto- and human-pathogenic bacteria [27,31], fungi are also highly sensitive to allicin [32]. Furthermore, virucidal effects of allicin have also been described [33]. In addition, it has been shown that various cancer cell lines are also killed by allicin [34]. Both protein thiols and low-molecular weight thiols, such as the tripeptide glutathione, can be oxidised by allicin to give *S*-thioallylated derivatives [35]. This reversible modification of thiol residues can lead to inactivation of essential enzymes or loss of structural protein function, and effects on cell-signalling pathways [35,36]. It was shown in human Jurkat cells, for example, that the cytoskeleton is a very important target for allicin and that the tubulin and actin cytoskeleton are destroyed by allicin, although tubulin appeared to be more sensitive than actin [37]. Not only in animal cells, but also in plant cells, was it shown that allicin destroyed the integrity of the cytoskeleton [38].

The effects of *S*-thioallylation have already been investigated in human cells as well as in pathogenic microorganisms [35,39]. In particular, the immune system is modulated by stimulation with allicin, which can be important, for example, in the course of infection with COVID-19 [40].

Allicin (*M*_r_ = 162) is not only active in solution, but also via the gas phase [32]. This is of course of great potential importance for inhalative applications, especially since a large number of potential lung pathogens are also effectively inhibited by allicin [41,42,43]. Accordingly, an in vitro model was developed to simulate flow conditions in the human lung from the second to fifth bronchial generation, which, by lining the model with an agar layer seeded with microorganisms, allows us to determine whether and where inhalative application of allicin might be effective in the treatment of pathogens [*ibid*]. Previous work was done with bacterial pathogens. Our approach here with fungi provides important preliminary data before potentially progressing to testing in animals. The aim of this study was the development of a test rig for use with a suitable non-pathogenic hyphal fungus which grows at 37 °C and the confirmation of the potential application of allicin for treatment of pulmonary mycoses in an in vitro model.

## 2. Results

### 2.1. Identification and Molecular Characterisation of a Suitable Thermophilic Fungus

A total of three fungi were isolated from the environment and morphologically and molecularly characterised (Appendix B). The isolated species were *Penicillium crustosum*, *Cladosporium cladosporioides* and *Paecilomyces formosus*. Only *P. formosus* showed growth at 37 °C and was thus a candidate for further investigation in the lung test rig.

### 2.2. Allicin Inhibits Spore Germination upon Direct Contact and via the Gas Phase

Susceptibility of filamentous fungi to 40 µL aqueous allicin solution (10 mM resp. 20 mM) was observed in an agar-diffusion test after 48 h of incubation. Triplicates were made and deionised water (diH_2_O) was used as a control. Inhibition zone diameters were measured. Mean inhibition zone diameter and the standard deviation (*SD*) were calculated. Using 10 mM aqueous allicin solution, a 14 mm diameter inhibition zone formed in the case of *P. crustosum*; in the presence of 20 mM allicin, they had an average size of 16.7 mm (*SD* = 0.6) (Figure 1). Mean inhibition zone diameter of *C. cladosporioides* using 10 mM allicin solution was 36 mm (*SD* = 6) and 40 mm (*SD* = 5) in the presence of 20 mM allicin. Mean inhibition zone diameter of *P. formosus* with 10 mM allicin was also 40 mm. The largest inhibition zones were 47 mm (*SD* = 1) in the presence of 20 mM allicin for *P. formosus*.

The effectiveness of allicin in inhibiting fungal growth via the gas phase was also investigated. Therefore, an inverted inhibition zone test with the sample solution applied to the lid of the Petri dish was performed in triplicate. Inhibition zone diameter was measured. No growth inhibition was observed on plates treated with deionised water as well as 96% ethanol (Figure 2). 50 mM allicin caused a reduction and inhibition of spore germination at application volumes of 20 µL as well as 40 µL. *P. crustosum* showed the least growth inhibition. Although areas with low growth occurred in the region above the application point of the allicin solution, no clearly defined edge to the inhibition zones could be identified. Thus, no inhibition zone diameters could be accurately determined. In the case of *C. cladosporioides* and *P. formosus*, clear areas without fungal growth were discernible. A concentration-dependent allicin effect was also determined. The average inhibition zone diameter occurring for *C. cladosporioides* when treated with 20 µL 50 mM allicin was 41.3 mm (*SD* = 2.1); *P. formosus* had an average inhibition zone diameter of 49.7 mm (*SD* = 1.2) with the same treatment. Application of 40 µL 50 mM allicin resulted in larger inhibition zones for both fungi. Here, the mean inhibition zone diameter of *P. formosus* was 57 mm (*SD* = 1.0) and 47.5 mm (*SD* = 1.3) for *C. cladosporioides*.

### 2.3. Comparison of Allicin Efficacy with That of Amphotericin B

In the following, the antifungal effectiveness of allicin was compared with that of the commercially available antifungal agent amphotericin B (ampB) (Figure 3). Therefore, triplicates were made and the mean effective concentration (EC_50_), at which only 50% of the spores germinate, and the minimum inhibitory concentration (MIC) were determined. In this context, the MIC describes the lowest concentration that must be applied to completely inhibit spore germination.

First, the effectiveness of amphotericin B in inhibiting spore germination was tested. AmpB concentrations tested were lower than those of allicin in general. *P. crustosum* had the highest EC_50_ associated with ampB with 42 µM compared to that of *C. cladosporioides* (21 µM) and *P. formosus* (3 µM). *P. formosus* showed the highest susceptibility, even a 6.75 µM ampB solution was sufficient to completely inhibit spore germination. For *C. cladosporioides*, the MIC was 54 µM and for *P. crustosum* 108 µM. Regarding allicin, *P. crustosum* was least susceptible which is supported by an EC_50_ of approx. 129 µM and a MIC value of 250 µM. In contrast, for *C. cladosporioides*, only about one ninth of the allicin concentration was needed to inhibit 50% of the spores (EC_50_ = 12 µM). Complete inhibition also occurred much earlier, an allicin concentration of 63 µM was sufficient to completely inhibit spore germination. Similar effective concentrations were found for *P. formosus*: EC_50_ was 24 µM and MIC 63 µM.

### 2.4. Allicin Aerosol Inhibits Fungal Growth in an Artificial Lung Model

For aerosol treatments, the lung model, coated with agar embedded *Paecilomyces* spores, was treated in a temperature-controlled incubator box (T = [37 ± 1] °C) with different concentrations of allicin dissolved in deionised water. Starting from a 50 mM stock solution, allicin concentrations of 0.1 mM, 0.25 mM, 0.5 mM and 0.75 mM were prepared. The model was also treated with pure deionised water and 96% ethanol aerosol. Initial treatment volume was 8 mL. One treatment of each triplicate is shown. When treated with deionised water, spores germinated uniformly and fungal growth throughout the model was observed (Figure 4A). The 96% ethanol aerosol inhibited spore germination completely. At 0.1 mM allicin, a slight inhibition on the bronchial surfaces at the carinal branch points was detected. At 0.25 mM allicin, a clear inhibition in these areas was observed. The effect was strengthened by 0.5 mM allicin aerosol. With 0.75 mM allicin aerosol, growth inhibition of allicin-susceptible *P. formosus* was complete on all the bronchial surfaces throughout the model. Binary images created with GIMP allowed quantification of overgrown (black pixels) and bare (white pixels) areas in the lung model (Figure 4B) as well as statistical analysis (Figure 4C). Compared to the water control, there is a concentration-dependent, statistically significant reduction in fungal growth in the lung model that underwent allicin aerosol treatment.

### 2.5. Enhancement of Allicin Vapour Efficacy Using Ethanol as Solvent

Lung model experiments with allicin vapour were performed analogously to the aerosol experiments, but the aerosol generator and the associated nebuliser vessel were not required. Thus, with the air flow only gaseous components of the test solution were transported into the lung model. Treatment with deionised water did not inhibit fungal growth (Figure 5A). In contrast to the aerosol treatment, 96% ethanol showed a lower antimycotic effect via the gas phase. Fungal growth was detectable, particularly in the second and fourth bronchial generations. Less intense growth was detectable throughout the model. Treatment with 20 mM aqueous allicin solution caused a minor growth reduction. The surfaces of the model along the first bifurcation, as well as the last bronchial generations, showed less severe blue staining and consequently lower growth. In contrast, 40 mM aqueous allicin solution caused almost complete inhibition of fungal growth. Using 96% ethanol as solvent, 20 mM allicin solution almost completely inhibited fungal growth. Allicin efficacy via the gas phase was as effective in 20 mM allicin solution with ethanol as solvent was in aqueous 40 mM allicin solution. These macroscopically visible findings were confirmed by generating binary images, quantification of overgrown (black pixels) and bare (white pixels) areas and statistical analysis (Figure 5B,C).

Lastly, the amounts of allicin consumed in the treatments described above were calculated (Table 1). Therefore, allicin concentration used was multiplied by the solution volume consumed. To observe antifungal activity, higher initial concentrations were needed for vapour treatments. Weak antimycotic activity was observed after the treatments with low concentrated allicin aerosol. A concentration-dependent increase in antifungal activity was observed. More interesting are the observations related to vapour treatment. When 20 mM allicin solution is used, more allicin is carried into the model using 96% ethanol as solvent. This results from an increase in treatment volume. Mean treatment volume of 1.379 mL with ethanol as solvent is about five times higher than with water as solvent. As a result, the amount of allicin entering the lung model is also 5× higher. This calculation was confirmed by HPLC measurement with comparable results.

## 3. Discussion

It has been shown in various previous studies that allicin is well suited to killing lung pathogenic bacteria, either via the gas phase or as an aerosol. In order to get an overview of the deposition behaviour of allicin in vitro, a lung simulation apparatus was developed and tested with bacteria [42].

However, it is also known that bacteria are apparently more resistant to allicin than fungi [44]. One may speculate whether this is related to the cytoskeleton. In this context, hyphae coming into contact with allicin showed morphological abnormalities [45]. Although further experimental studies are required to explain this differential susceptibility, an investigation of the effect against potential fungal pulmonary pathogens seems interesting considering the fact that pulmonary mycoses are increasingly becoming a problem. Hence, there is an urgent need for new active agents due to the worsening resistance problem [46,47,48].

To conduct these experimental studies, natural fungal strains potentially capable of surviving at a temperature of 37 °C, and therefore viable as possible infectious agents, were first isolated from environmental samples and subsequently taxonomically identified using morphological and molecular techniques. Three species were successfully isolated. Three primer pairs were used for PCR amplification, which was considered sufficient for a reliable species determination after sequencing. In addition, microscopic analysis was performed (Appendix B), which, with the help of the appropriate identification literature, led to the same result as molecular analysis, providing a very reliable identification result. *Penicillium crustosum*, as with almost all representatives of the genus *Penicillium*, is largely known as a plant-pathogen or saprophyte. Therefore, this fungus is in principle suitable for a proof-of-principle but cannot be used to understand human pathogenic situations outside of in vitro investigations. *Cladosporium cladosporioides* is a widespread melanised fungus that lives primarily saprophytically in a wide variety of substrates, especially soil. In addition to its widespread role as a trigger of allergies, *C. cladosporioides* has also been identified in clinical preparations in humans and animals. Pulmonary infections with *C. cladosporioides* have also been observed and the immune response to them has been studied in the experimental mouse model [49]. Thus, *C. cladosporioides* represents a potential, albeit rare, human pathogen. Finally, *Paecilomyces formosus* belongs to a group of fungi that are phylogenetically and morphologically very close to the genus *Penicillium*. Similar to their close relatives, representatives of the genus *Paecilomyces* mostly have a saprophytic lifestyle. Nevertheless, there are also descriptions of infections with *P. formosus*, for example in the way that a chronic granulomatous disease was due to an infection with *P. formosus* [50]. However, even with this fungus, which has been known as a human pathogen since 1963, mycoses have occasionally been observed, primarily in immunosuppressed or immunocompromised patients [51]. Accordingly, *P. formosus* is a very good experimental organism to perform the in vitro experiments with.

As a first step, the susceptibility of the different fungal isolates to allicin was tested. As a first variant, spores were embedded in agar and susceptibility to allicin was tested qualitatively in a plate diffusion test. While *P. crustosum* shows a comparatively low susceptibility to allicin (recognisable by the small inhibition zone), *C. cladosporioides* and *P. formosus* are similarly sensitive and show a large inhibition zone (Figure 1). It should be kept in mind that this assay, as with the following one, examines spore germination and thus does not examine the effect on already existing hyphae. The following test examines the effect on spore germination in a quantitative method. Here, the rate of germinated spores is examined as a function of the concentration of the active substance and thus a minimum inhibitory concentration of the active substance can be determined. Allicin is compared with the widely used fungicide amphotericin B (Figure 3). Amphotericin B is effective against all three isolates, although the sensitivity expressed in the EC_50_ value differs considerably. While *P. crustosum* has an EC_50_ of 42 µM and is thus relatively resistant to amphotericin B, the EC_50_ value of *C. cladosporioides* is half as low at 21 µM. *P. formosus* is considerably more sensitive with 3 µM (Figure 3). *P. crustosum* is also least sensitive to allicin; the EC_50_ value here is 119 µM. *C. cladosporioides* is the most sensitive with 12 µM and thus more sensitive to allicin than to amphotericin B. *P. formosus*, on the other hand, possesses an EC_50_ value of 24 µM which is also well within the range of amphotericin B. This indicates that allicin as a natural substance is similarly as effective as the pharmaceutically used substance amphotericin B, and it is demonstrated that allicin has potential as a fungicide (Figure 3).

*Penicillium* shows considerable resistance to allicin, since there are also a large number of garlic-inhabiting *Penicillium* isolates. Most likely an adaptation to the defence substances of garlic has been developed, as shown in the bacterial system using a *Pseudomonas fluorescens* isolate (*Pf*AR-1) [28]. The further investigation of allicin resistance in *Penicillium* could be an interesting approach to understanding eukaryotic allicin resistance. However, *C. cladosporioides* and *P. formosus* are also sensitive to allicin in the gas phase (Figure 2). Above the allicin-containing droplet, a clear inhibition zone is shown, although there was no direct contact between allicin and the fungus. This indicates that only efficacy via the gas phase is possible.

Volatile antibiotics and antimycotics are of the utmost interest, especially for use in the lung. They allow direct application from the air side of the mucosa and enable direct treatment without the need for bronchioalveolar lavage with an antibiotic-containing solution during pneumonia treatment [52]. Note that fluid mechanics becomes necessary to investigate the deposition of the active substance on the lung epithelium. We compare vapour treatment to an application with aerosols, which are of course much easier to evaluate mechanically due to describable particle size, as we have already done for the model used here in earlier work. Although the complexity of the experimental approach is reduced by a simplified lung model, it allows for the determination of the fundamental efficacy of the substances. The lung model is lined with agar in which fungal spores are incorporated. This model is then treated with allicin aerosol (Figure 4) or allicin vapour (Figure 5) and the evaluation takes place after 24 and 48 h. This is to check whether a short-term inhibition after 24 h is not wrongly interpreted as killing. It can be stated that lower concentrations of allicin are required for treatment with aerosols due to direct contact. The amount of 0.5 mM of allicin inhibits the fungus such that no overgrowth can be observed even after 48 h, which is certainly still the case at lower concentrations (Figure 4). In the gas phase treatment, in contrast to the aerosol treatment, higher initial concentrations are needed. These are only half as high when allicin is dissolved in ethanol as when it is dissolved in water (20 mM vs. 40 mM). It appears that ethanol is a very well-suited carrier substance that promotes the transition of allicin into the gas phase. Although the concentrations used were higher in the allicin vapour treatments, this form of treatment offers application advantages. While allicin aerosols may deposit and cause local tissue burns, respiratory tract cell damage could be reduced by inhalation of allicin vapour. Experiments with lung cells need to be conducted.

In summary, allicin shows good efficacy against fungi both in direct contact and via the gas phase. Previous studies have shown that fungi are generally more susceptible to allicin than bacteria [44]. It is assumed that this is due to the cytoskeleton. According to the current available data, this must be considered as a primary target for allicin action. However, further research is needed to prove this assumption. In principle, however, allicin is very effective on fungi, such that on the basis of the present in vitro investigations, it can be assumed that an application for the treatment of pulmonary mycoses could be promising, which must undoubtedly be tested in further experiments, also in vivo.

## 4. Materials and Methods

### 4.1. Allicin Synthesis

Allicin synthesis was performed, with slight modifications, according to Albrecht et al. [53]. Instead of chromatographic separation, hexane (Carl Roth, Karlsruhe, Germany) and dichloromethane (Carl Roth, Karlsruhe, Germany) extraction was used.

### 4.2. Isolation and Characterisation of Fungi

Fungal samples were isolated from the environmental samples to obtain natural strains that are ubiquitously distributed and to illustrate the widespread potential risk of infection. Growth was described macroscopically on different culture media (Appendix A) and microscopically. Species identification was performed using the DNA barcoding method combining PCR amplification and sequencing of nuclear ribosomal internal transcribed spacer 1 (*ITS1*) locus, the nuclear beta-tubulin (*TUB2*) gene and the nuclear calmodulin (*CALM*) (Table 2). Primers were obtained from Integrated DNA Technologies (Leuven, Belgium).

### 4.3. Sequencing and Species Identification

PCR amplicons were sent to the Microsynth Seqlab sequencing laboratory in Göttingen (Germany) for sequencing. Sequencing results were compared with the NIH GenBank^®^ using the Basic Local Alignment Search Tool (BLAST^®^, Bethesda, MD, USA) and with the ARS Microbial Genomic Sequence Database (USDA Agricultural Research Service, https://data.nal.usda.gov/dataset/ars-microbial-genomic-sequence-database-server, accessed 9 September 2020).

### 4.4. Cultivation Methods

If not described differently, identified *Penicillium crustosum* and *Cladosporium cladosporioides* were routinely cultivated at room temperature on PDA plates for 3 to 5 days. Similarly, identified *Paecilomyces formosus* was routinely cultivated at 37 °C on PDA plates for 3 to 5 days.

### 4.5. Susceptibility Assays

Spore suspensions from a 3- to 5-day-old pure plate culture were prepared by flooding the plate with water and transferring 10 mL into a sterile centrifuge tube to fill it up to 50 mL with water. Spore count of 10 µL suspension was determined by using a Thoma counting chamber. A sufficient amount of spore suspension was added to 20 mL molten PDA medium at 50 °C to obtain a spore concentration of 10^6^ spores per mL. The mixture was poured immediately into a Petri dish to make spore-seeded agar plates. For the agar diffusion test, wells (Ø = 0.6 cm) were punched out of the solidified agar with a cork borer to apply 40 µL allicin solution. To investigate antifungal activity of allicin vapour, different amounts of 50 mM allicin solution (20 µL and 40 µL) were pipetted onto the Petri dish lid, and the solidified agar plate with fungal spores was placed inverted over the lid. Fungal growth was evaluated after 48 h incubation.

### 4.6. MIC and EC_50_ Determination

Two-power dilution series of allicin in water (0.3–325 µg/mL = 1.95 µM–2 mM) and amphotericin in 3% DMSO (0.1–99.8 µg/mL = 0.11–108 µM) were prepared. Spore suspensions were prepared as they were for the inhibition zone tests. 50 µL of a test solution was mixed with 50 µL of spore suspension (final concentration: 2 × 10^4^ spores per mL) in a well of a 96-well microtiter plate. Plates were covered with air-permeable, self-adhesive cling film (Carl Roth, Karlsruhe, Germany) and incubated for 48 h without shaking. Spore germination was studied under a microscope and the ratio of germinated spores to total spore number determined. The lowest concentration without spore germination gave the MIC. EC_50_ indicates the concentration at which 50% of the spores did not germinate.

### 4.7. Lung Model Experiments

The construction details of the lung model, the general experimental setup and implementation of a treatment have been previously reported [42,43]. The model consists of two halves and represents the 2nd to 5th generation (trachea = 0th generation) of average life-size bronchial passages in a human lung. Coating it with agar medium simulates the epithelial surface of the bronchi and allows the incorporation of fungal spores to simulate an infected lung.

The casting process of the bronchial surface coating of spore-containing agar was performed using a negative mould with a positive stamp with an offset of 1 mm. Before the agar surface was cast, the surfaces were sterilised with UV light for 20 min. The 2.0% (*w*/*v*) PDA medium was autoclaved for 20 min at 121 °C and subsequently tempered in a water bath at 50 °C prior to use. Spore suspension of *Paecilomyces formosus*, as prepared for susceptibility testing, was added to a 20 mL aliquot of tempered agar to obtain a concentration of 10^6^ spores per mL. The mixture was rapidly filled into the mould and the stamp was immediately pushed down by the precise positioning of dowel pins. The stamp was held for 5 min to ensure complete agar solidification. Afterwards, the stamp was removed and excess medium at the inlet and outlets was trimmed off with a scalpel to obtain sharp inlet and outlet contours. This procedure was repeated for the other model half and both halves were assembled to yield the model bronchial system.

The assembled model was placed in the incubation chamber and attached to the aerosol generator (TurboBOY SX with PARI LC SPRINT nebuliser nozzle attachment, Pari GmbH, Starnberg, Germany) and air supply. The nebuliser was attached to the compressor, an external air-supply and pressure gauge. Total airflow entering the lung model via the long inlet tube was set to 6 L per min. The inlet tube connected seamlessly with the agar-coated lung model and ensured a fully developed laminar airflow into the model. Temperature in the incubation chamber was regulated by a copper heating plate with aluminium cooling fins attached and on–off settings at 37 ± 1 °C. Air was circulated within the chamber by means of a fan placed 1 cm from the heating plate. A 1.5 cm diameter air outlet from the incubation chamber was fitted with a 0.45 µm filter.

For aerosol treatment, allicin solutions diluted in deionised water (diH_2_O) were used. For the treatment with allicin vapour, allicin was diluted either in deionised water or 96% ethanol and placed into a glass to yield a 5 cm^2^ surface for evaporation at room temperature. The residual volume of the allicin solution was then measured and the quantity of allicin entering the lung model during the treatment was calculated from the treatment volume (difference of volume used and residual).

After discontinuation of the airflow, the assembled lung model was covered with moist filter paper, wrapped in aluminium foil and placed in an incubator for incubation at 37 °C for 24 h and 48 h to allow fungal growth. The model was dismantled and separated, exposing the inner surfaces. The agar layer was carefully removed from the model and placed on a Petri dish. Each half was sprayed with 0.5% 3-(4,5-dimethylthiazol-2-yl)-2,5-diphenyltetrazolium bromide (MTT) solution, covered with transparent polyethylene foil and incubated at 37 °C for 2.5 h. Metabolically-active fungal cells reduced the MTT to dark colored formazan, and where fungal growth was inhibited the agar remained pale. Fungal growth on the lower half of the model was photographically documented. The photo’s background has been removed using removebg (https://www.remove.bg/de/upload, accessed 27 December 2021) to subsequently create binary images and count the pixels using the GNU Image Manipulation Program (GIMP, Version 2.10.30, The GIMP Development Team).

### 4.8. Statistical Analysis

Statistical analyses were performed using SigmaStat, Version 3.11.0 (Systat Software, San José, CA, USA). A *p* value ≤ 0.05 was considered statistically significant. Statistical significance of susceptibility tests was calculated by One-Way RM ANOVA using the Holm–Šidák method. For lung model aerosol experiments, treatments were compared to water control and statistical significance was calculated in a Student’s *t*-test. For lung model vapour experiments, differences between allicin treatments were examined and statistical significance was calculated in a Student’s *t*-test.

## Figures and Tables

**Figure 1 ijms-23-06607-f001:**
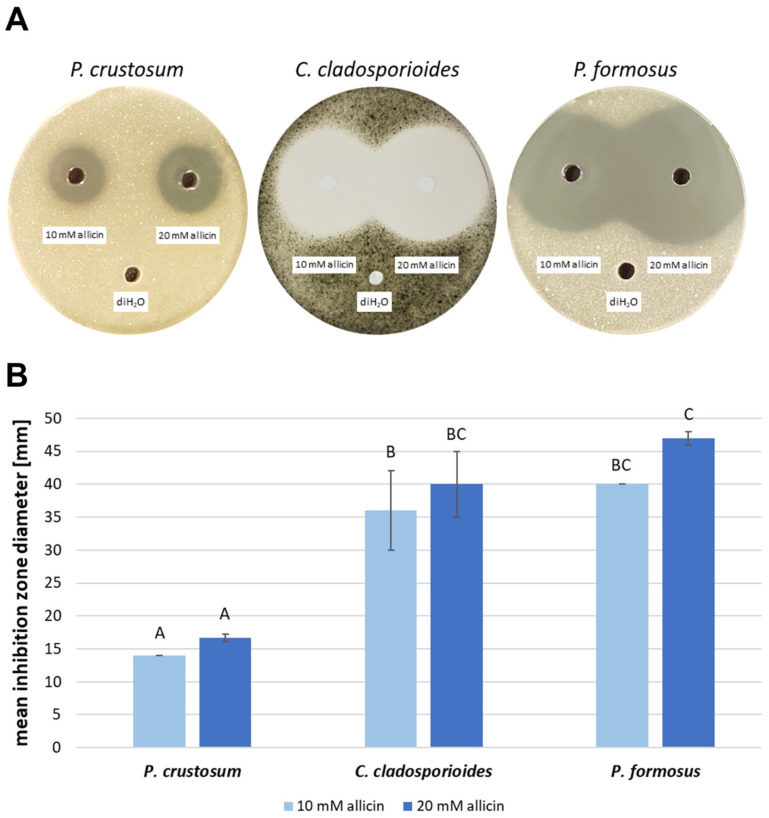
Allicin inhibits spore germination upon direct contact in a concentration-dependent manner. (**A**) Fungal spores (10^6^ spores per mL) were added to 20 mL PDA medium (50 °C). After solidification, three holes (Ø = 0.6 cm) were punched into the agar and filled with 40 µL of 10 µM, 20 µM aqueous allicin solution or deionised water (diH_2_O), respectively. Triplicates were made for each fungus and incubated for 48 h. (**B**) *n* = 3, error bars show standard deviation. Same letter indicates no significant difference (*p* > 0.05) in a One-Way RM ANOVA using the Holm–Šidák method.

**Figure 2 ijms-23-06607-f002:**
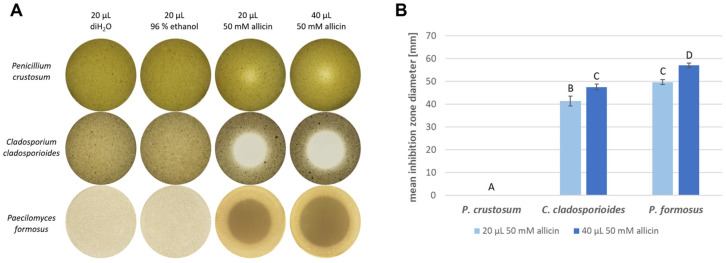
Allicin inhibits spore germination via the gas phase. (**A**) Fungal spores (10^6^ spores per mL) were added to 20 mL PDA medium (50 °C). After the medium solidified, diH_2_O, 96% ethanol or 50 mM aqueous allicin solution were applied to the Petri dish lid. Triplicates of each fungal sample were prepared and incubated for 48 h. (**B**) *n* = 3, error bars show standard deviation. Same letter indicates no significant difference (*p* > 0.05) in a One-Way RM ANOVA using the Holm–Šidák method.

**Figure 3 ijms-23-06607-f003:**
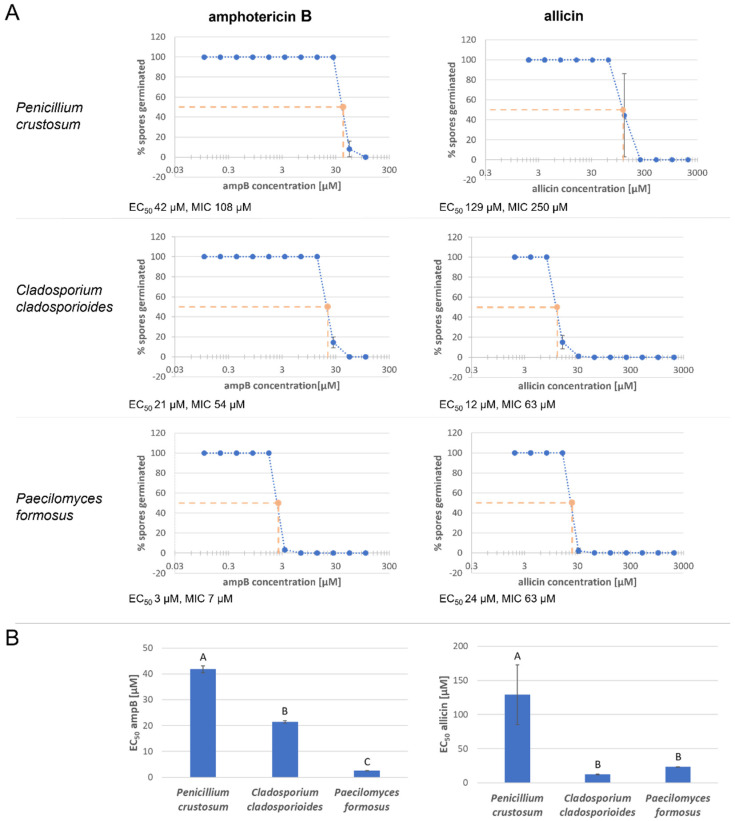
Allicin efficacy is comparable to that of amphotericin B. Fungal spore suspensions were mixed with sample solution. Spore germination was examined microscopically after 48 h of incubation. (**A**) Germinated and ungerminated spores were counted in an image field and the ratio of germinated spores to total spore number was calculated (diH_2_O or 3% DMSO was set to 100% spore germination). The mean values of three biological replicates with three technical replicates each are shown. Sample solution concentration was plotted logarithmically. Error bars represent the standard deviation. (**B**) *n* = 9, error bars show standard deviation. Same letter indicates no significant difference (*p* > 0.05) in a One-Way RM ANOVA using the Holm–Šidák method.

**Figure 4 ijms-23-06607-f004:**
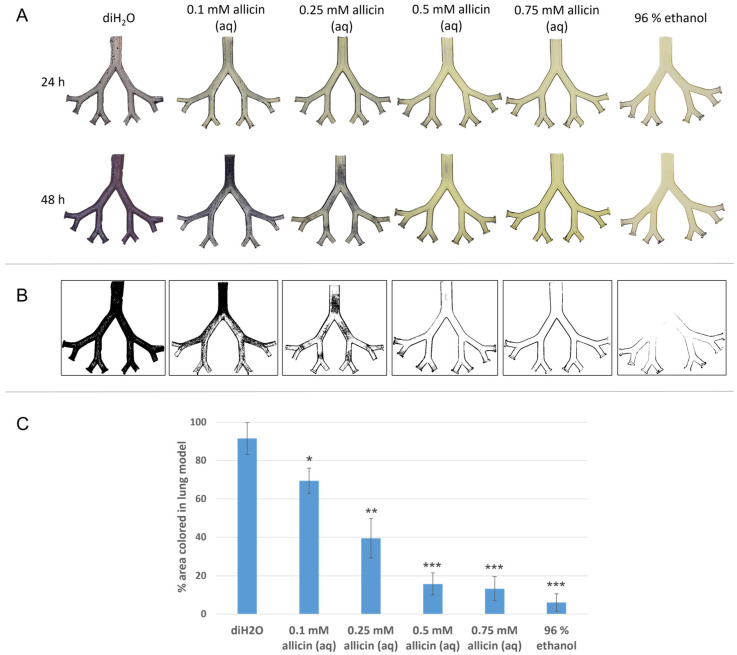
Concentration-dependent inhibition of spore germination after aerosol treatment of the lung model with aqueous allicin solution. A volume of 20 mL of 2% (*w*/*v*) PDA medium (50 °C) mixed with spore suspension of *Paecilomyces formosus* (10^6^ spores per mL) was used to coat each lung model half-shell. The supply air was adjusted to 6 L per min. The model was treated with aqueous allicin solution in the lung simulation apparatus for 20 min and subsequently incubated at 37 °C. (**A**) After 24 h and 48 h of incubation, fungal growth was detected by staining with MTT. The lower half of the experiment model is shown. (**B**) Binary images of the 48 h incubated model halves. (**C**) *n* = 3, error bars show standard deviation. Significant differences to the water control in a Student’s t-test are marked by asterisks (*p* < 0.05 = *; *p* < 0.01 = **; *p* < 0.001 = ***).

**Figure 5 ijms-23-06607-f005:**
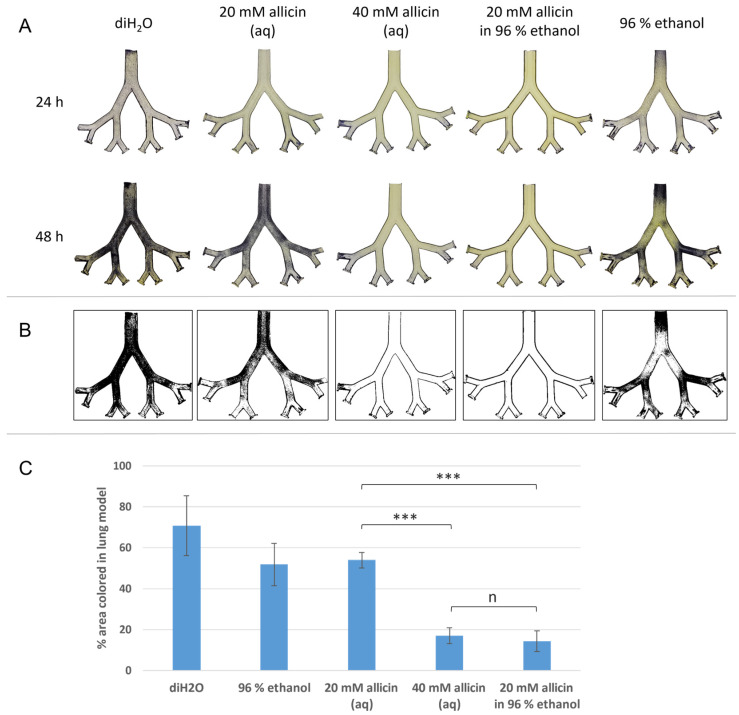
Ethanol as a solvent promotes the antimycotic activity of allicin via the gas phase. For gas phase experiments, synthetic allicin was diluted in either deionised water (diH_2_O) or 96% ethanol. The lung model was coated the same as for the aerosol treatment (20 mL 2% (*w*/*v*) PDA with 10^6^ spores per mL of *Paecilomyces formosus* per half). The supply air was adjusted to 6 L per min, treatment duration was 20 min. The model was subsequently incubated at 37 °C. (**A**) After 24 h and 48 h of incubation, fungal growth was detected by staining with MTT. The lower half of the experiment model is shown. (**B**) Binary images of the 48 h incubated model halves. (**C**) *n* = 3, error bars show standard deviation. Significant differences between the treatments in a Student’s t-test are marked by asterisks (*p* < 0.001 = ***; n = non-significant).

**Table 1 ijms-23-06607-t001:** Calculation of allicin consumption in the treatments. *n* = 3, antifungal activity is displayed and is color-coded: red = weak, yellow = medium, green = strong.

Treatment	Aerosol				Vapour		
Allicin concentration [mM]	0.1	0.25	0.5	0.75	20	40	20
Solvent	deionised water				deionised water		96% ethanol
Mean treatment volume [mL]	4.581 (±0.020)	4.974 (±0.041)	5.001 (±0.073)	4.935 (±0.175)	0.273 (±0.033)	0.454 (±0.021)	1.379 (±0.003)
Mean allicin consumed [µmol]	0.46 (±0.002)	1.2 (±0.01)	2.5 (±0.04)	3.7 (±0.13)	5.5 (±0.7)	18 (±1.4)	28 (±0.1)

**Table 2 ijms-23-06607-t002:** Primers used for amplification and sequencing of marker genes to determine fungal species.

Amplicon	Primer Name	Primer Sequence (5′-3′)	Source
** *ITS1* **	ITS1F	CTT GGT CAT TTA GAG GAA GTA A	[54]
	ITS2	GCT GCG TTC TTC ATC GAT GC	
** *TUB2* **	TUB2Fd	GTB CAC CTY CAR ACC GGY CAR TG	[55]
	TUB4Rd	CCR GAY TGR CCR AAR ACR AAG TTG TC	
** *CALM* **	CAL-228F	GAG TTC AAG GAG GCC TTC TCC C	[55]
	CAL-737R	CAT CTT TCT GGC CAT CAT GG	

## Data Availability

The data that support the findings of this study are available from the corresponding author upon reasonable request.

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
