# Peer review of "Allicin as a Volatile or Nebulisable Antimycotic for the Treatment of Pulmonary Mycoses: In Vitro Studies Using a Lung Flow Test Rig"

_ijms, 2022, doi:10.3390/ijms23126607_

Round 1

Reviewer 1 Report

 - In terms of language and narrative, the article seems to be correctly written and it is easy to follow the arguments, experimental and results. It is appropriately illustrated and, as far as I know, no typos or spelling errors were noticed.

- The article itself is interesting because as the authors state and we all can agree, that due to increasing resistance of pathogenic organisms, in this case, fungi, to current treatments, “there is a need for novel antimycotics” (line 73). In this work the authors show a series of different experiments showing each one allicin as effective inhibitor against three different fungi.  As a result the authors state that for allicin “it can be assumed that an application for the treatment of pulmonary mycoses could be promising” (line 251). And again, we can actually agree with this.

-However there are some details that may need some explanation:

            - It is quite strange nobody before has studied the well-known allicin as potential antimycotic compound. But, well, that is what it is and the authors make an initial study of this.

            - The authors explain in lines 265-290 that the choice of the three fungi were based on the fact that they are capable of surviving at temperature of 37 oC. However, none of them are actually found as fully pulmonary infectious fungi though one of them, C. cladosporioides has been observed in pulmonary infections (line 279) and P. formosus, have been observed as human pathogen “primarily in immunosuppressed or immunocompromised patients” (line 288). The authors quite honest and declare that “Penicillium crustosum is in principle suitable for a

proof-of-principle but cannot be used to understand human pathogenic” (line 272-275). In any case, I do not see any problem with this. The experiments are well designed, show clear results and can be perfectly used as a starting point for future studies for actual lung mycotic infections.

            - Depending on the tests, allicin is delivered in different ways:  as direct contact, disc difussion (section 2.2), as aerosol (section 2.4) or as vapour (section 2.5). In these cases, 96% ethanol is used as blank or as allicin carrier. According to disc diffusion experiment in line 138, 96% ethanol showed no fungi growth inhibition. However, in the aerosol experiment the authors claim 96% ethanol inhibited spore germination completely (line 192). Later for the vapour test, 96% ethanol “showed a lower antimycotic effect via the gas phase” (line 217). In fact, this solvent is used as ideal allicin carrier compared to water and the authors claim that “Ethanol as a solvent promotes the antimycotic activity of allicin via the gas phase” (line 230 on Figure 5. Captions). This needs some clarification.  Is 96% ethanol actually inhibiting fungi or not? If there is promotion of antimycotic activity when ethanol is the solvent, would not be possible that the solvent itself is cause of this activity as in the case of aerosol test?

In summary, the article is a good example of “proof-of-principle” as the authors mention, for this antifungal study. This opens another application for allicin and as also mentioned, there can be further experiments in vivo.

I would approve the publication of this manuscript if somehow the ethanol issue is explained.

Reviewer 2 Report

The effect of allicin on the inhibition of fungal lung infections under airflow conditions mimicking the human lung was investigated. In my opinion, the manuscript is well written. The methodology was properly planned. The results were correctly described and interpreted. Correct conclusions were formulated.
However, in my opinion, the authors should cite more new publications in the work, i.e. from 2020-2022.

Author Response

We would like to thank you for your positive and constructive feedback. Indeed, we were happy to take up the suggestion and cite more up-to-date literature (see MS), which certainly means an improvement of the manuscript. Thank you for the advice!

Reviewer 3 Report

After reading the manuscript, I have no comments and I believe in my opinion that it can be adopted in the current version as it is. The quoted article is a very valuable scientific item in supplementing the literature data that is niche in this field. Congratulations to the authors. My sincere appreciation to them.

Author Response

We would like to thank you very much for the positive and encouraging feedback on our MS.